# 5-hydroxymethylcytosine marks regions with reduced mutation frequency in human DNA

**Marketa Tomkova, Michael McClellan, Skirmantas Kriaucionis\*, Benjamin Schuster-Boeckler\***

Ludwig Cancer Research Oxford, University of Oxford, Oxford, United Kingdom

**Abstract** CpG dinucleotides are the main mutational hot-spot in most cancers. The characteristic elevated C>T mutation rate in CpG sites has been related to 5-methylcytosine (5mC), an epigenetically modified base which resides in CpGs and plays a role in transcription silencing. In brain nearly a third of 5mCs have recently been found to exist in the form of 5-hydroxymethylcytosine (5hmC), yet the effect of 5hmC on mutational processes is still poorly understood. Here we show that 5hmC is associated with an up to 53% decrease in the frequency of C>T mutations in a CpG context compared to 5mC. Tissue specific 5hmC patterns in brain, kidney and blood correlate with lower regional CpG>T mutation frequency in cancers originating in the respective tissues. Together our data reveal global and opposing effects of the two most common cytosine modifications on the frequency of cancer causing somatic mutations in different cell types.

**\*For correspondence:** skirmantas. kriaucionis@ludwig.ox.ac.uk (SK); benjamin.schuster-boeckler@ ludwig.ox.ac.uk (BS-B)

**Competing interests:** The authors declare that no competing interests exist.

## Introduction

Cancer genomics projects have revealed that the distribution of somatic mutations across the genome is not uniform (*Lawrence et al., 2013*). Apart from positive and negative selective pressure, a number of factors can influence regional mutation frequencies, such as chromatin organisation (*Schuster-Böckler and Lehner, 2012*), replication timing (*Koren et al., 2012*), metabolic load (*Ames et al., 1993*) and exposure to different mutagens (*Poon et al., 2013*). Furthermore, highly transcribed regions generally exhibit lower mutation frequencies due to the influence of transcription-coupled repair (*Lawrence et al., 2013*). In addition to the regional distribution of mutations, the local nucleotide contexts and mutation types (referred to as *mutational signatures*) have been investigated extensively since they provide critical clues about the mechanism of mutagenesis. For example, consensus motifs for cytidine deaminases (such as APOBEC and AID) were found enriched at mutational hot-spots, suggesting that activity of these enzymes could be the potential cause of those mutations (*Nik-Zainal et al., 2012*; *Taylor et al., 2013*). The most frequent mutational signature found in the majority of cancers is C to T transition in a CpG dinucleotide context (CpG>T) (*Alexandrov et al., 2013*; *Lawrence et al., 2013*). This relates to the fact that cytosines in CpG dinucleotides are frequently methylated to form 5-methylcytosine (5mC). The rate of spontaneous deamination of 5mC into T is four fold higher than the rate of deamination of C into U (*Lindahl and Nyberg, 1974*). In the germline of vertebrates, this likely facilitated a general depletion of CpGs outside of CpG islands which are largely unmethylated.

The genomes of all examined vertebrate species feature DNA methylation, and loss of DNA methylation is incompatible with normal development in mice (*Li et al., 1992*; *Okano et al., 1999*). DNA methylation plays a role in gene expression, most notably by repressing one allele of imprinted genes. Moreover, it is involved in maintenance of genome stability, alternative splicing, X chromosome inactivation and suppression of retrotransposons (*Klose and Bird, 2006*; *Jones, 2012*). In

**eLife digest** A molecule called DNA encodes genetic information inside our cells. Random changes to the DNA sequence, known as mutations, can occur in any cell. Most mutations are harmless, but some can lead to disease – most prominently cancer. Like how car accidents can happen more often on some roads than others, mutations are more frequent in some parts of the DNA. Cytosine, one of the four letters of the genetic code, usually accumulates more mutations than the other three letters.

Cytosine can be decorated with distinct 'marks' to form either methyl-cytosine or hydroxymethyl-cytosine. Methyl-cytosine is known to mutate relatively easily, and is the most common type of mutation observed in most cancers. However, little was known about how easily hydroxymethyl-cytosine mutates.

Modifications of cytosine are distributed differently in cells from different tissues. To test whether hydroxymethyl-cytosine mutates more or less often than methyl-cytosine in human cells, Tomkova et al. used the cytosine mutations measured in human brain, kidney and blood cancer samples. Comparing these mutations to maps of cytosine modifications from healthy tissues of the same type revealed that in all tissues, hydroxymethyl-cytosine appears to mutate less often than methyl-cytosine.

There are several possible explanations for the difference in mutation frequency between methyl-cytosine and hydroxymethyl-cytosine. Tomkova et al. plan to investigate these possibilities further in an effort to fully understand the underlying mechanisms that drive cytosine to mutate.

2009, 5-hydroxymethylcytosine (5hmC) was indisputably shown to exist in DNA of brain and other tissues (*Kriaucionis and Heintz, 2009*). It was concurrently shown that ten-eleven translocation (TET) enzymes are able to convert 5mC into 5hmC (*Tahiliani et al., 2009*). Unlike 5mC, which is observed at similar levels in many cell types, the abundance of 5hmC varies widely. 5hmC was observed to be particularly enriched in brain cells (*Kriaucionis and Heintz, 2009*; *Lister et al., 2013*) and detectable in embryonic stem cells and all examined tissues (*Tahiliani et al., 2009*; *Globisch et al., 2010*; *Szwagierczak et al., 2010*; *Wu and Zhang, 2011*). 5hmC and higher oxidised states of methyl-cytosine have been proposed to play a role in de-methylation via ineffective re-methylation after replication or directly by thymine DNA glycosylase (TDG) (*Tahiliani et al., 2009*; *He et al., 2011*; *Maiti and Drohat, 2011*; *Shen et al., 2013*; *Hu et al., 2014*). In addition to demethylation, 5hmC has been implicated in transcriptional regulation, and a number of DNA binding proteins recognising 5hmC have been identified (*Mellén et al., 2012*; *Spruijt et al., 2013*; *Takai et al., 2014*). 5hmC is found depleted in primary tumours and TET2 is frequently mutated in myelodysplastic syndrome, acute myelogenous leukaemia and T-cell lymphoma, indicating that 5hmC plays a role in carcinogenesis. However, the molecular mechanism by which 5hmC affects carcinogenesis is poorly understood (*Rasmussen and Helin, 2016*).

5hmC is an important intermediate during demethylation in zygotes and ES cells (*Tahiliani et al., 2009*; *Inoue and Zhang, 2011*; *Wossidlo et al., 2011*), but the vast majority of 5hmC is found as a stable, long-lived modification in adult mouse tissue that undergoes little cell division (*Bachman et al., 2014*; *Brazauskas and Kriaucionis, 2014*). Thus, we hypothesised that – similar to 5mC – long-lived 5hmC could have a substantial influence on the mutability of DNA. Little is known about the mutational properties of 5hmC, in part because until recently there has been a lack of information on the precise location of 5hmC in the genome. With the development of techniques for single-nucleotide resolution mapping of 5mC and 5hmC (*Yu et al., 2012*; *Booth et al., 2014*), it is now possible to differentiate mutation rates at 5mC and 5hmC sites. Recently, Supek et al. (*Supek et al., 2014*) reported elevated C>G transversion rates at 5hmC sites, using 5hmC maps from human and mouse embryonic stem cells. However, these findings are limited by the fact that embryonic stem cells differ substantially from the somatic tissues in which mutations were observed (*Schultz et al., 2015*).

A large proportion of mutations observed in any cancer genome originate in its pre-cancerous cell of origin (*Nik-Zainal et al., 2012*; *Stephens et al., 2012*; *Tomasetti et al., 2013*; *Wu et al.,*

*2015*) and will have been influenced by its epigenetic landscape. The publication of single-base resolution maps of 5mC and 5hmC occupancy in samples of human brain, kidney and blood (*Wen et al., 2014*; *Chen et al., 2015*; *Pacis et al., 2015*) now enables us to interrogate the tissue-specific effect of cytosine modifications on somatic mutation rates in unprecedented detail.

Since 5hmC has been shown to be most abundant in human brain (*Li and Liu, 2011*; *Nestor et al., 2012*), we have initially focussed on assessing the relationship between mutability and DNA modifications in brain cancers. Based on a DNA sequencing data from five brain cancer types encompassing 665 patients, we show that the dominant mutational signature in brain cancers is CpG>T, which is modulated by the modification state of cytosine. Strikingly, the CpG>T mutation frequency of 5-hydroxymethylcytosine is reduced nearly two-fold compared to the methylated state. We find that the ratio of 5hmC to 5mC in 100 kb genomic intervals correlates with CpG>T mutation frequency even after accounting for confounding factors like gene density or CpG islands. When we expand our analysis to include mutations and 5hmC maps from kidney and myeloid lineage of blood cells, we observe a clear tissue-specific effect of 5hmC on mutagenicity. Finally, we measured 5mC and 5hmC levels using methodology of high accuracy in eight different human tissue types and show that reduced 5hmC levels associate with an increased proportion of CpG>T mutations in cancers of the corresponding tissue. Together, our findings suggest that hydroxymethylation has a significant influence on the likelihood of mutations at CpG sites across many human tissue types.

## Results

We compiled base-resolution maps of 5mC and 5hmC frequency in brain, kidney and myeloid cells from publicly available sources (*Wen et al., 2014*; *Chen et al., 2015*; *Pacis et al., 2015*). All three data sets are based on bisulfite (BS) and 'Tet-assisted bisulfite' (TAB) sequencing, respectively. BS-Seq detects any modified cytosine (i.e., does not distinguish 5mC and 5hmC) whereas TAB-Seq specifically detects 5hmC. The combination of the two methods allows an estimation of the levels of both 5mC and 5hmC for all sufficiently covered cytosines. As 5hmC predominantly occurs in a CpG context, we focussed the analysis on CpG sites. Sequencing reads come from heterogeneous populations of cells. Hence, a single locus usually cannot be assigned a single state (C, 5mC or 5hmC). Instead, we estimated the frequency of modification, hydroxymethylation and methylation per site using the percentage of BS-Seq reads that showed a modification (referred to as *mod level*), the percentage of TAB-Seq reads that showed hydroxymethylation (referred to as *5hmC level*) and their difference (*5mC level = mod level – 5hmC level*), respectively.

### 5hmC sites in brain exhibit lower frequency of CpG>T mutations than 5mC sites

Since brain tissue has been shown to exhibit particularly high levels of 5hmC (*Figure 1A*), we first investigated the relationship between the regional distribution of 5hmC, 5mC and mutagenesis in brain tumours. We reasoned that this approach would provide the highest sensitivity to detect any correlation between 5hmC and mutation frequency.

We analysed 344370 somatic single nucleotide variants (SNVs) from 665 samples derived from exome and whole genome sequencing of the following cancer types: Glioblastoma (GBM), Glioma low grade (GLG), Neuroblastoma (NRB), Medulloblastoma (MDB) and Pilocytic astrocytoma (PA) (*Alexandrov et al., 2013*). The dominant point mutation type in these cancers were C>T transitions in a CpG context (*Figure 1B,C*), similar to what was observed in combined analyses of all cancer types (*Alexandrov et al., 2013*).

Mutations and DNA modifications are not distributed uniformly along the chromosomes. Strikingly, 5hmC levels were visibly and significantly anti-correlated with the frequency of CpG>T (r=-0.71, chr3), while 5mC levels displayed a positive correlation (r=0.66, chr3, *Figure 1D*, *Figure 1—figure supplements 1–2*). This is not a simple consequence of the uneven distribution of genes, exons, CpG islands or levels of gene expression (*Figure 1—figure supplement 3* and additional analyses below).

Averaging over the entire genome, the frequency of C>T mutations differed substantially between $5mC_{high}$ ($5mC_{high}$: *mod level* > 10% and *5hmC level / mod level* $\leq$ 0.3) and $5hmC_{high}$ ($5hmC_{high}$: *mod level* > 10% and *5hmC level / mod level* $\geq$ 0.5) sites. The fraction of mutated $5hmC_{high}$ sites was significantly lower than the fraction of mutated $5mC_{high}$ sites (*Figure 1E*). The

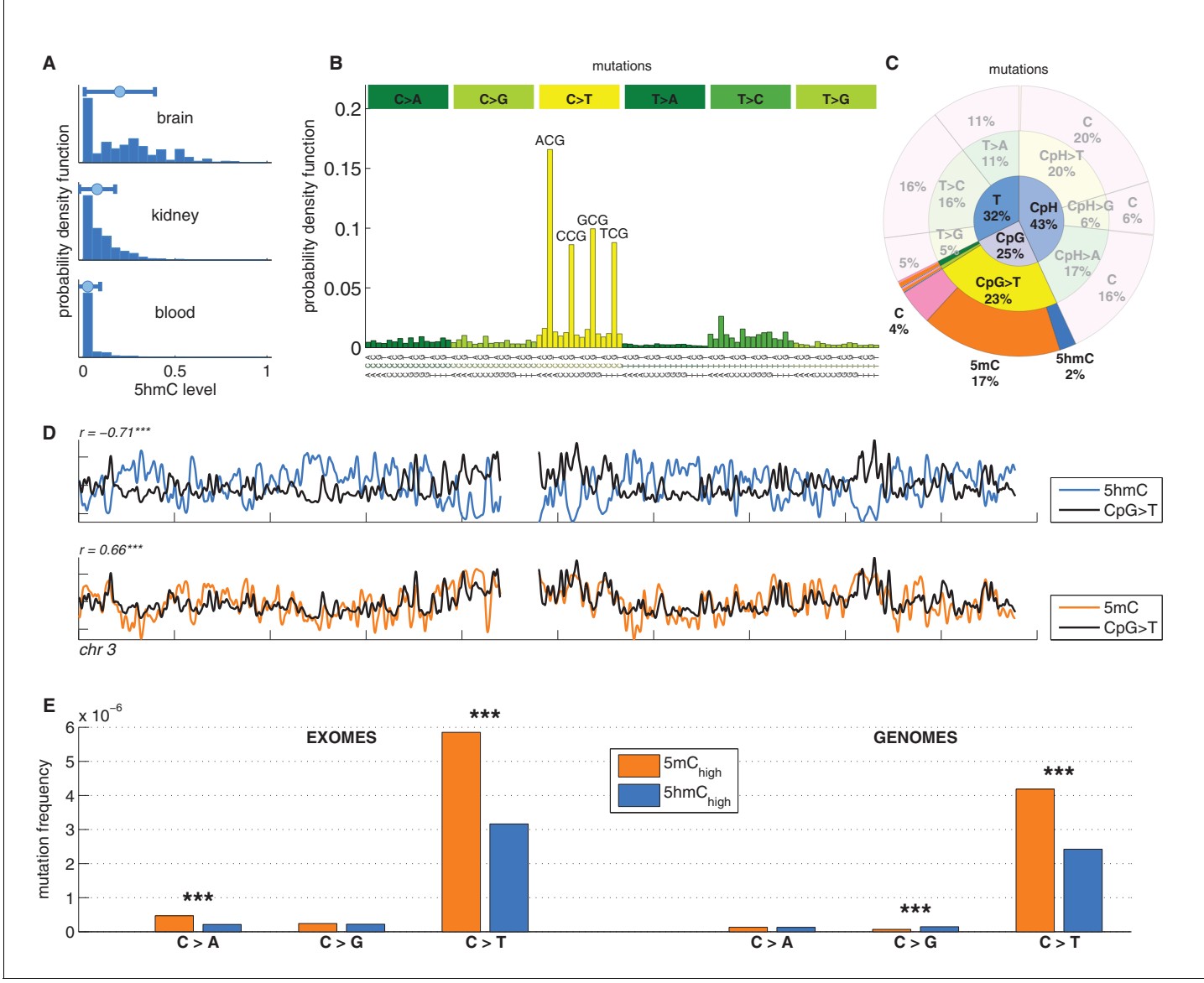

**Figure 1.** C>T mutations are common in the genome but depleted in 5hmC sites compared to 5mC sites. (**A**) Distribution of 5hmC in a CpG context in brain compared to kidney and blood. (**B**) Frequency of SNVs in brain cancer exomes, stratified by sequence context, normalised by frequency of trinucleotides. (**C**) Distribution of single-nucleotide variants (whole genomes) in brain cancer according to type, context and modification state. (**D**) CpG>T mutation frequency (black), 5hmC (blue) and 5mC (orange) density in 100 kbp windows of chromosome 3, smoothed with a Gaussian filter (n = 50, sigma = 2.5). (**E**) Average fraction of mutated sites for $5mC_{high}$ vs. $5hmC_{high}$ over all patient samples (CpG sites only; ***p<0.001; **p< 0.01; *p< 0.05, see Materials and methods).

The following figure supplements are available for figure 1:

**Figure supplement 1.** Distribution of CpG>T mutations vs modifications across all chromosomes.

**Figure supplement 2.** Distribution of CpG>T mutations vs modifications across all chromosomes.

**Figure supplement 3.** Distribution of CpG>T mutations vs other genomic features.

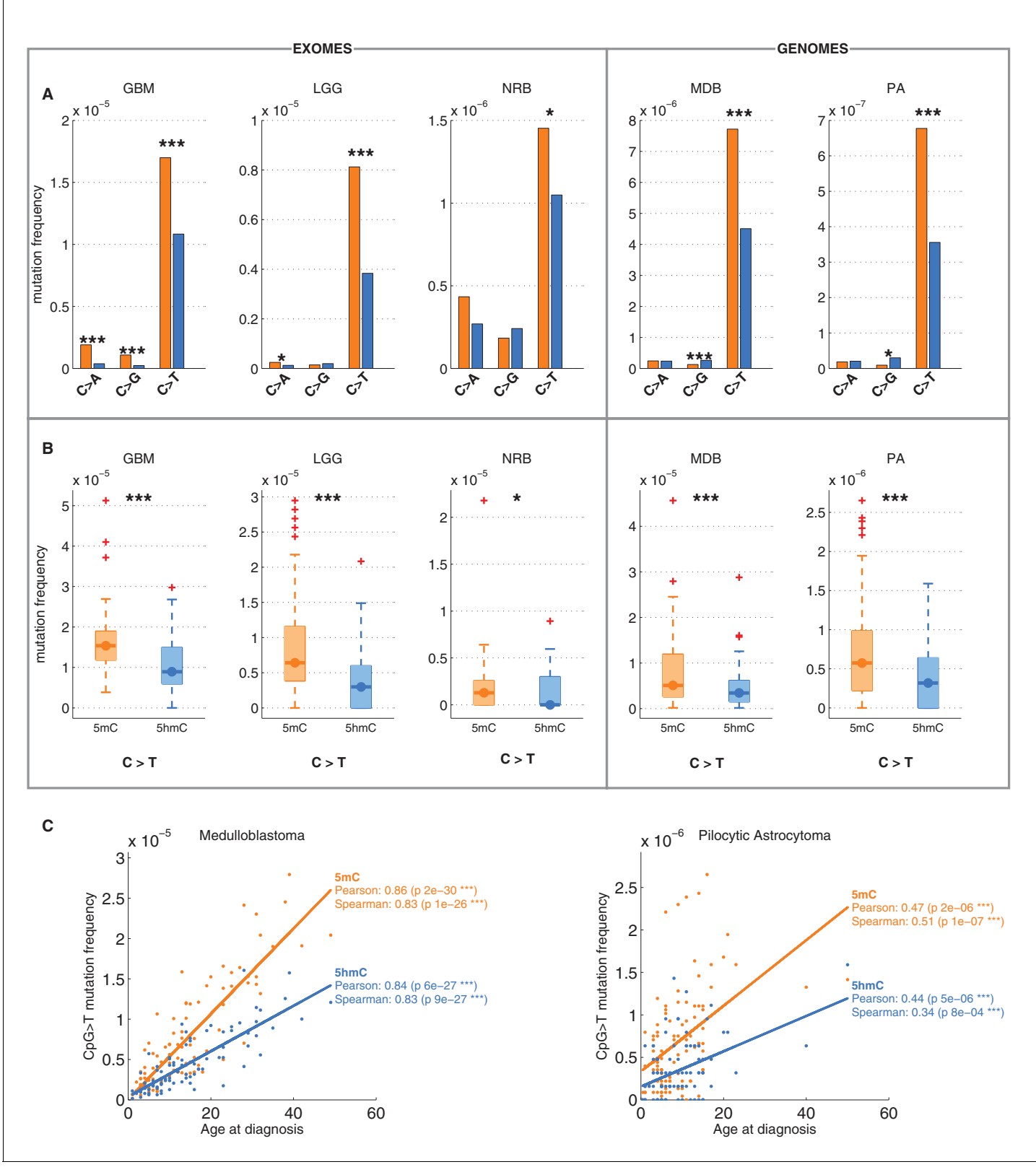

**Figure 2.** Differential mutation frequency between 5mC and 5hmC is present in all 5 brain cancer types and correlates with age at diagnosis. (A) Average fraction of mutated CpG sites for 5mC_high vs. 5hmC_high computed separately for each cancer type. (B) Box plot of C>T mutation frequency, as shown in A. (C) Correlation of whole genome CpG>T mutation frequency with age at the time of diagnosis in patients with Medulloblastoma and Pilocytic Astrocytoma.

*Figure 2 continued on next page*

*Figure 2 continued*

The following figure supplement is available for figure 2:

**Figure supplement 1.** Depletion of C>T mutations in 5hmC$_{high}$ is relatively insensitive to varying definitions of 5mC$_{high}$ and 5hmC$_{high}$.

lower mutation frequency was consistently observed in data derived from both exome and whole genome sequencing projects (p<0.001, Wilcoxon signed-rank test). Moreover, all brain cancer types individually displayed a significant (28–53%, p<0.05 in all types) reduction of C>T mutations in 5hmC$_{high}$ sites (*Figure 2A,B*).

It has been shown that CpG>T mutations are one of the two mutational signatures correlating with age (*Alexandrov et al., 2015*), supporting the fact that these mutations were gathered during the entire lives of the patients, not only after the origin of cancer. Here we observed that this correlation is present in both methylated and hydroxymethylated sites (*Figure 2C*). Moreover, the slope for 5mC was steeper than for 5hmC, suggesting that even the mechanisms causing the difference of CpG>T mutability between 5mC and 5hmC were present in the pre-cancerous cell of origin.

We also compared the fraction of mutated 5mC$_{high}$ and 5hmC$_{high}$ sites for the other two possible types of mutations: C>A and C>G. As shown in *Figure 1E*, C>A or C>G transversions are an order of magnitude less frequent than C>T transitions in both 5mC and 5hmC sites. The relationship between C>A and C>G mutations and 5hmC varied between cancer types. In GBM and LGG the frequency of C>A mutations was significantly higher in 5mC$_{high}$ compared to 5hmC$_{high}$ sites, but in NRB, MDB and PA we detected no significant difference. The frequency of C>G mutations in 5mC$_{high}$ sites differed significantly from 5hmC$_{high}$ sites only in MDB, PA and GBM. In MDB and PA, 5hmC$_{high}$ sites were slightly enriched for C>G mutations, whereas in GBM an enrichment was observed at 5mC$_{high}$ sites. Since C>T transitions are the most common somatic mutation type in brain and the difference in C>T mutations between 5mC$_{high}$ and 5hmC$_{high}$ sites is more consistent among cancer types than in the C>A and C>G transversions, we focus mainly on C>T mutations in the remainder of this report.

We confirmed that C>T mutations are significantly depleted at 5hmC sites across a wide range of thresholds in definitions of 5mC$_{high}$ and 5hmC$_{high}$ (*Figure 2—figure supplement 1A–F*). In fact, more stringent definitions of 5hmC (e.g., 5hmC$_{high}$: *5hmC level / mod level* $\geq$ 0.7) result in even greater differences (42–59%) in C>T mutation frequencies between 5mC$_{high}$ and 5hmC$_{high}$ sites (*Figure 2—figure supplement 1G–I*, *Figure 3—figure supplement 1A–D*), but these definitions would reduce the number of sites too much for our subsequent statistical analyses.

## Reduced 5hmC mutability in brain is not accounted for by genomic regions or gene expression

We next examined whether the decreased frequency of C>T transitions at 5hmC vs. 5mC sites might be an indirect effect of 5hmC being associated with genomic regions of lower mutability. Levels of 5mC and 5hmC vary across genomic regions. For example, 5hmC density is elevated in highly expressed genes in brain (*Mellén et al., 2012*; *Yu et al., 2012*; *Lister et al., 2013*; *Wen et al., 2014*). The observed decrease in C>T mutation frequencies might therefore be attributable to higher gene expression, which would correlate with higher transcription coupled repair. We therefore performed the analysis described above separately for regions with high vs. low gene expression in human brain (see Materials and methods). There was a lower overall mutation frequency in highly expressed genes (*Figure 3A–B*), but both highly and lowly expressed genes exhibited significantly lower C>T transition rates at 5hmC sites compared to 5mC sites (*Figure 3A–D*). This suggests that the observed effect is independent of transcription and thus not a result of transcription coupled repair.

Gene expression is only one of many possible region-related confounding factors. Hence, to correct for any regional variation, we performed the analysis on groups of sites generated by pairing the modified CpGs: each 5hmC site was paired with the nearest yet unpaired 5mC site from an equivalent genomic and sequence context (an approach adapted from *Supek et al., 2014*, see Materials and methods). Thereby we compared the mutation frequencies of two groups (one group comprising 5mC sites and one group comprising 5hmC sites) containing the same number of loci

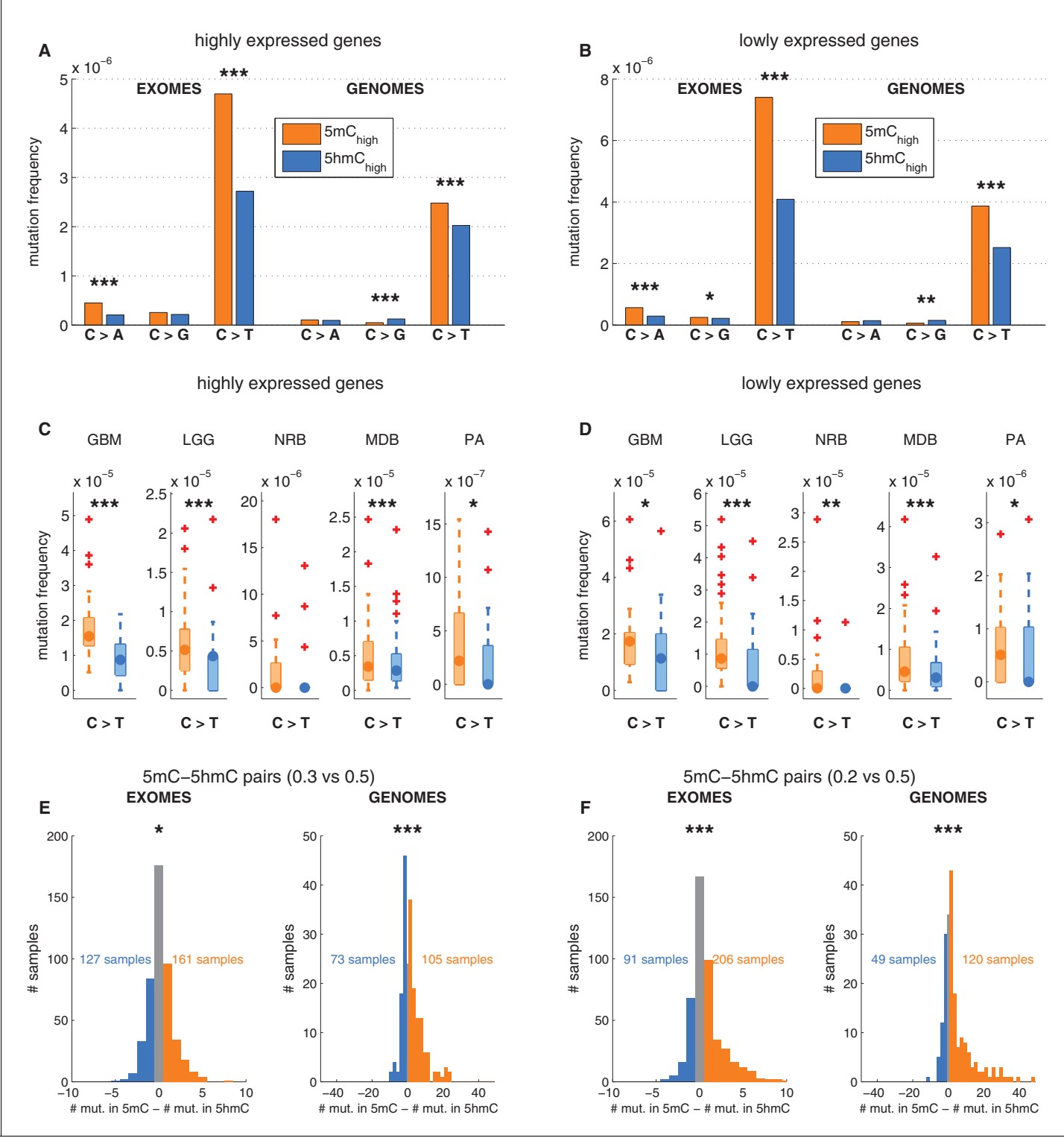

**Figure 3.** Depletion of C>T mutations in 5hmC sites is not explained by gene expression or regional mutation rate variation. (A–B) Frequency of mutations in $5mC_{high}$ vs $5hmC_{high}$ sites within highly expressed (A) or lowly expressed (B) genes (see Materials and methods). (C–D) Boxplot visualisation of C>T mutation frequency for each cancer type. (E) For each patient sample, the overall difference in mutations in paired sites was calculated and compared using a Wilcoxon signed-rank test. Shown here is a histogram of samples by the difference in mutations for paired 5mC and 5hmC sites (negative values shown blue, positive in orange; see Materials and methods for details). Mutations in 5mC sites exceed paired 5hmC sites, causing a shift to the right. (F) Same as E but using a more stringent definition of 5mC (only sites with $threshold_{5mC} \leq 0.2$).

*Figure 3 continued on next page*

*Figure 3 continued*

The following figure supplement is available for figure 3:

**Figure supplement 1.** Depletion of C>T mutations in 5hmC$_{high}$ is relatively insensitive to varying definitions of 5mC$_{high}$ and 5hmC$_{high}$.

(6801374 cytosines in each group). As a result of this experimental setup, a substantial fraction of mutated 5mC sites were excluded, greatly reducing the statistical power of this 'paired' analysis. Nevertheless, the frequency of C>T mutations in 5hmC remained significantly lower than in 5mC in both exomes and genomes (*Figure 3E–F*), supporting that the difference between 5mC and 5hmC mutation frequency is not caused by regional differences.

To ensure that there is no confounding bias in the spatial distribution of mutations around 5mC or 5hmC sites, respectively, we plotted mutation frequencies in a 2 kb radius up and downstream of modified loci (*Figure 3—figure supplement 1G*, Materials and methods). The mutation frequency differed substantially in the aligned positions of DNA modifications but was indistinguishable in the surrounding area. In conclusion, regional mutation rate variability is unlikely to account for the difference in C>T mutational load in 5mC and 5hmC sites.

## Relative 5hmC correlates with CpG>T mutation frequency

The 5mC and 5hmC frequency at each base reflect the prevalence of each modification within the sequenced cell population. We hypothesised that if 5hmC had a direct effect on C>T mutation likelihood, we would observe an increase in mutation frequency with decreasing 5hmC occupancy. To test this, we formally defined 5hmC$_{rel}$ as the ratio of the proportion of reads supporting 5hmC, relative to the proportion of reads supporting any modification at a particular cytosine (5hmC$_{rel}$ = *5hmC level / mod level*). We then divided brain CpG sites into bins according to their 5hmC$_{rel}$ level and computed the fraction of mutated sites in each bin (*Figure 4A*). We observed a clear linear relationship between 5hmC$_{rel}$ values and C>T mutation frequencies. Notably, the correlation was present in all the tested brain cancer types in exome- and whole genome-sequenced samples. A regression slope test confirmed significance of this relationship in all the cancer types. To confirm that the results are not influenced by an uneven distribution of information across bins, we performed quantile binning so that each bin contains an approximately equal number of positions (see Materials and methods). The results of quantile bins were equivalent to evenly spaced bins (*Figure 4—figure supplement 1H*).

For comparison, we also evaluated the relationship between 5hmC$_{rel}$ and the frequency of C>A and C>G mutations (*Figure 4A*). Consistent with our previous results, an increase in 5hmC$_{rel}$ is associated with an increase in C>G mutations in whole genomes (from MDB and PA samples), but the relationship in other cancer types shows no significant trend. C>A mutations decrease with 5hmC$_{rel}$ levels in GBM but exhibit no significant signal in the remaining tumour types.

This result supports the conclusion that the decrease in C>T mutation frequency at 5hmC sites is not an artefact of our chosen definition of 5mC or 5hmC. Even more importantly, it supports the notion that this decrease is directly caused by the properties of these DNA modifications.

## Mutation load of 5hmC sites is similar to unmodified cytosines

The findings reported so far could be attributed to an elevated mutation rate in 5mC, to a lowered mutagenicity of 5hmC or a combination of the two. To investigate this question, we compared mutation frequencies at 5mC and 5hmC sites to that of unmodified cytosines. We divided all the sequenced CpG sites into 9x9 bins according to their levels of 5mC and 5hmC. We observed that the mutation frequency of unmodified cytosine is similar to 5hmC, whereas 5mC exhibited much higher mutation frequency (*Figure 4B*). Further, we calculated the mutation frequency distribution in sites that exhibited almost no methylation or almost no hydroxymethylation, respectively. When methylated sites are excluded, the mutation frequency does not show any significant trend with increasing levels of 5hmC (*Figure 4C*). Conversely, excluding hydroxymethylated sites leads to a significant gradient in mutation frequency with increasing levels of 5mC (*Figure 4D*). When only fully modified sites (*mod level* $\geq$ 90%) are taken into account, increasing levels of 5hmC (i.e., decreasing levels of 5mC) are associated with a significant decrease in C>T mutation frequency (*Figure 4E*).

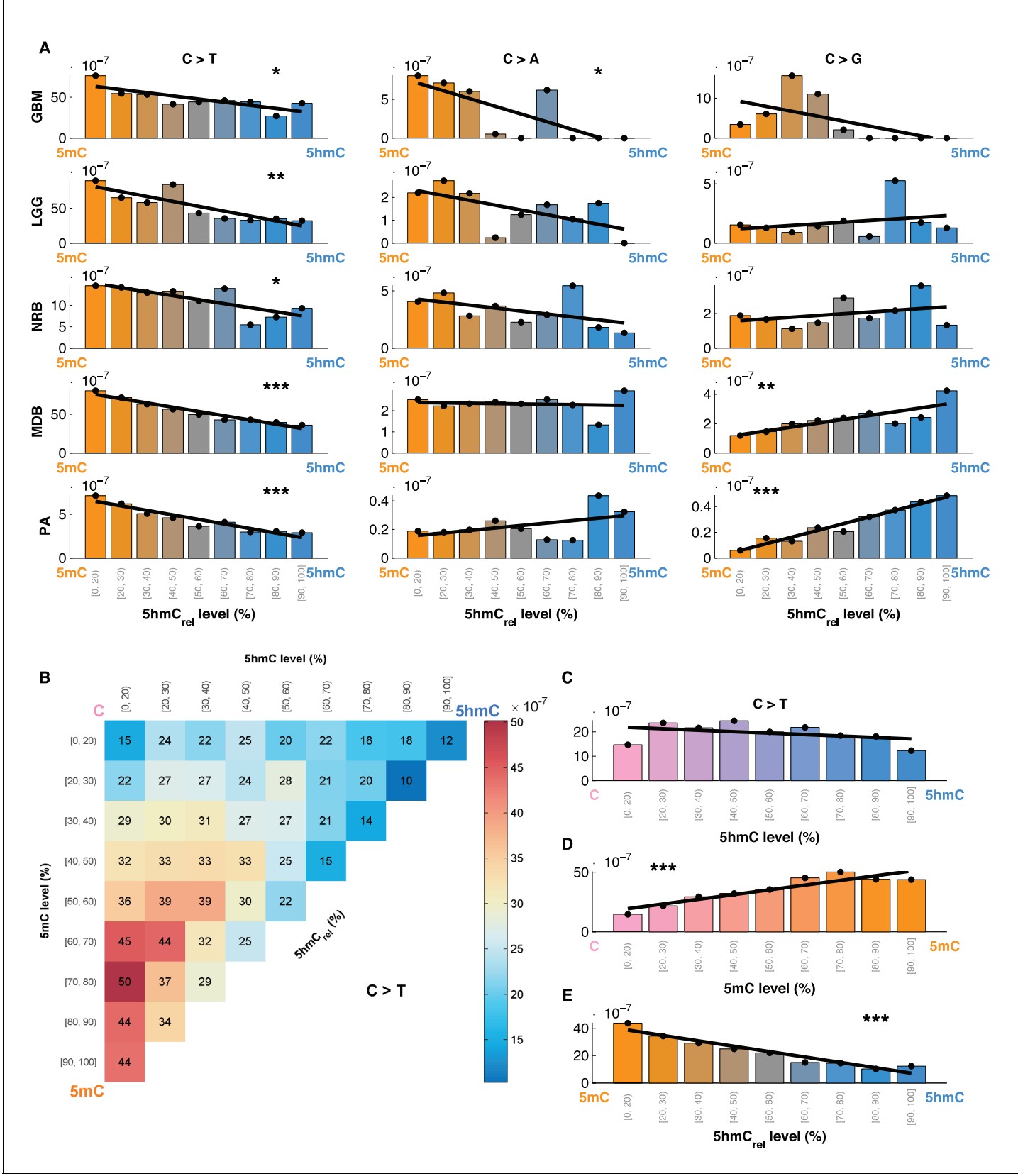

**Figure 4.** Mutation frequency negatively correlates with 5hmC_rel level per base. (**A**) Fraction of mutated CpG sites as a function of 5hmC_rel levels by mutation and cancer type. Bins to the left represent sites predominantly methylated, while bins to the right contain increasingly hydroxymethylated
*Figure 4 continued on next page*

*Figure 4 continued*

sites. Black line denotes linear regression fit (F-test for coefficient deviation from 0, see Materials and methods). (B) Distribution of CpG>T mutation frequency by modification type. The top left bin contains cytosines that are mostly unmodified, the bottom left bin contains exclusively methylated cytosines and the top right bin contains cytosines that are mostly hydroxymethylated. (C) Top row of B, *i.e.* distribution of mutations in unmethylated sites. (D) First column of B, *i.e.* distribution of mutations in sites without 5hmC. (E) Diagonal of B, i.e. distribution of mutations in highly modified sites.

The following figure supplement is available for figure 4:

**Figure supplement 1.** CpG>T mutation frequency as a function of 5hmC$_{rel}$ levels with equal binning (each bin contains approximately the same number of sites).

## 5hmC is a predictor of CpG>T mutation frequency across the genome

To examine the exclusive impact of DNA modifications on regional frequencies of mutations, we split the genome into 100 kb windows and fitted a generalised linear model to explain the observed per-window CpG>T mutation frequency from a combination of features including average 5mC and 5hmC levels, 5hmC$_{rel}$, gene density, CpG island density amongst others. Only whole genome sequencing data were used for this analysis. To compare the resulting models, we calculated their respective 'explained deviance' $D^2$, a generalisation of explained variance that is more appropriate for comparing generalised linear models (see Materials and methods).

The best individual predictor of CpG>T mutation frequency was 5hmC$_{rel}$ ($D^2$ = 0.11), outperforming all other features (*Figure 5A*). Interestingly, the sum of 5mC and 5hmC levels ('mod') performed worst, suggesting that bisulfite sequencing measurements alone are a poor predictor of mutagenicity. When combining all 11 features into one model, the total explained deviance for 100 kb windows was 16%.

Varying the chosen window size (3 kbp – 3 Mbp; *Figure 5B*, *Figure 5—figure supplement 1A–C*) does not substantially change the comparison of the predictive power of the respective features. In all cases, 5mC and 5hmC$_{rel}$ were the two best predictors, with 5hmC$_{rel}$ performing slightly better with smaller windows. However, the total explained deviance increased with window size, reaching values as high as 45% for univariate models and 60% for models with all predictors. This led us to question whether the increasing predictive power of larger windows has a biological reason, or whether it is a consequence of the lower data density in small windows.

Since many smaller windows contain no observed mutations, low $D^2$ values could simply reflect a lack of data. To test this, we generated simulated mutations so that a 'perfect' predictor was linearly related to the mutation likelihood per window (see Materials and methods). We then measured the effect of window and sample size (number of patients) on the observed $D^2$, repeating the simulations 10 times. The resulting curves of the explained deviance resemble those measured in the real data (*Figure 5—figure supplement 1D*). Moreover, in the simulated data, higher numbers of patients lead to higher $D^2$ even for smaller window sizes, suggesting that lower $D^2$ values in smaller windows indeed are a consequence of lower data density.

## Level of genic 5hmC correlates with decrease of CpG>T

It has been reported that 5hmC is enriched in gene bodies, and several brain cancer sequencing data sets. We therefore tested whether the relationship between 5hmC and mutations, which we observed across the whole genome, is also detectable in exonic regions alone.

In line with our earlier results, we found that 5hmC$_{rel}$ significantly contributes to the deviance explained by the model, beyond covariation with gene expression (*Figure 5C–D*; F-test p<2e-100). We hypothesised that this effect should be most pronounced when using modC>T and CpG>T as the response variable, whereas it should decrease when using definitions of mutations that include a progressively wider range of loci (C>T, C>N, N>N). Indeed, the unique contribution of 5hmC$_{rel}$ to the explained gene mutation frequency decreased as the mutation sets became larger and more distant from modC>T (*Figure 5C–D*, *Figure 5—figure supplements 2–3*). Nevertheless, in all of the cases, 5hmC$_{rel}$ significantly improved the fit of the model. Conversely, we confirmed that 5hmC$_{rel}$ had no significant predictive power for the frequency of T>N mutations (*Figure 5C–D*; column T>N), supporting the hypothesis that 5hmC$_{rel}$ selectively affects mutations in modified cytosines.

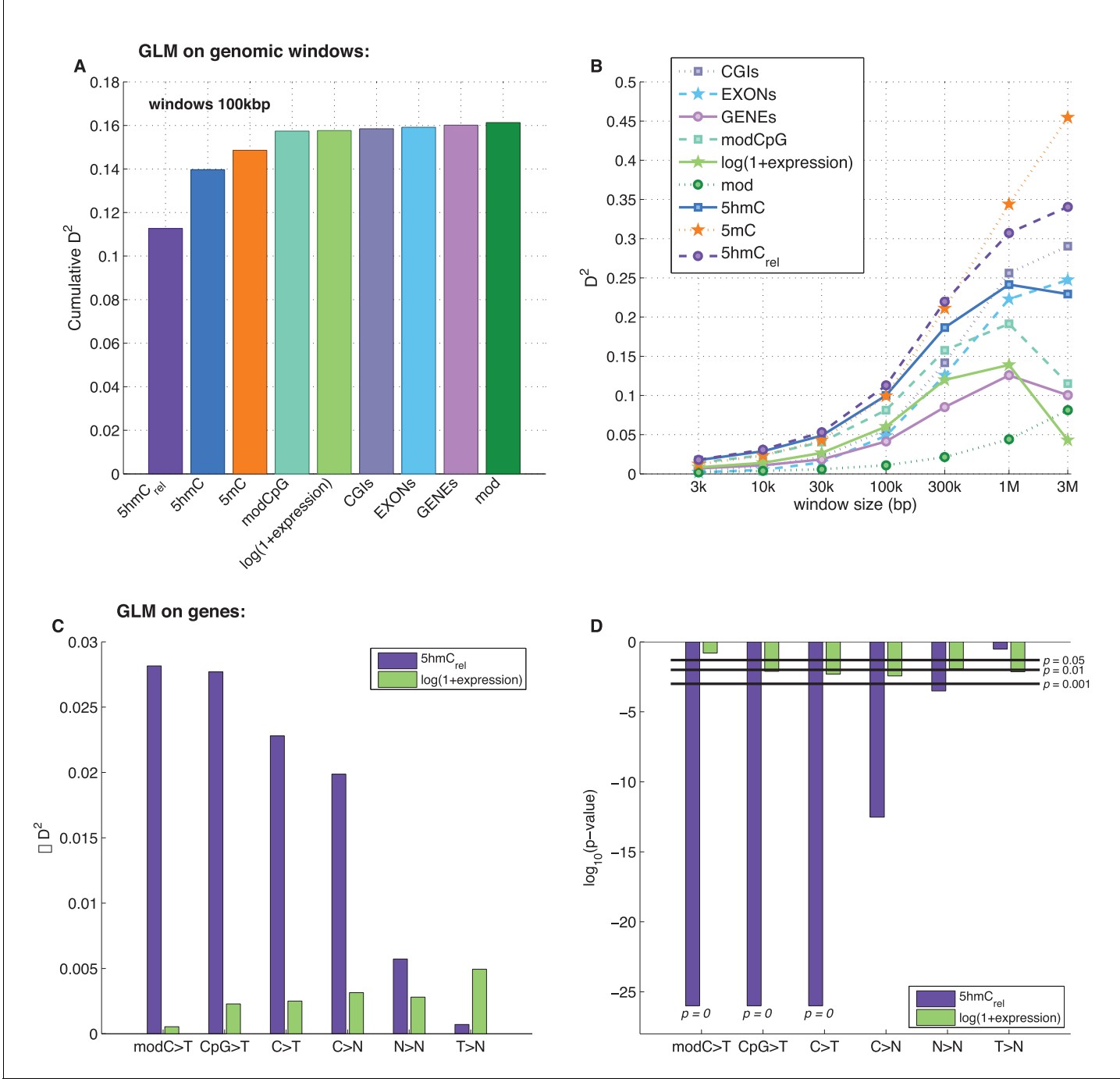

**Figure 5.** Predictors of mutations: 5hmC$_{rel}$ compared to other genomic features. (**A**) Prediction of CpG>T mutation frequency (using whole genome sequencing only) in 100 kbp genomic windows. Predictors are sorted according to the D$^2$ in a univariate model. The height of the $k$th bar denotes the D$^2$ of a model with the first $k$ predictors. (**B**) Comparison of the nine predictors of CpG>T mutation features by D$^2$ in a univariate models, in a range of window sizes. (**C**) Prediction of different types of mutation frequency in genes. Increase in D$^2$ of a generalised linear model including 5hmC$_{rel}$ over gene expression (purple) or gene expression over 5hmC$_{rel}$ (green) (see Materials and methods). (**D**) Significance of observations in **C** (see Materials and methods).

The following figure supplements are available for figure 5:

**Figure supplement 1.** Genome-wide prediction of CpG>T mutation frequency: 5hmC$_{rel}$ compared to other genomic features.

**Figure supplement 2.** Effects of 5hmC$_{rel}$ levels on gene mutability.

*Figure 5 continued on next page*

eLIFE Research article                                    Cancer biology | Computational and systems biology

Figure 5 continued

**Figure supplement 3.** Effects of 5hmC_{rel} levels on gene mutability.

## Decreased CpG>T mutation frequency in 5hmC is not limited to brain tissue

Two recently published datasets allowed us to address the question of mutational properties of 5mC and 5hmC also in two other tissues: kidney (*Chen et al., 2015*) and blood (*Pacis et al., 2015*). For blood we used 174 sequencing samples from Acute Myeloid Leukaemia (AML) as the cancer type closest to the blood dendritic cells in which the BS-Seq and TAB-Seq measurements were performed. For kidney we combined 585 samples from four distinct sequencing projects, covering Kidney Clear Cell, Kidney Papillary and Kidney Chromophobe carcinomas.

Matching our findings in brain, 5hmC sites were mutated significantly less frequently than 5mC sites in both tissue types (*Figure 6B*), irrespective of whether genome or exome sequencing data were used. Moreover, a similar difference was present in all available replicates of the BS-Seq and TAB-Seq measurements (6 for blood, 2 for kidney, *Figure 6—figure supplement 1A*).

Genomic distribution of 5hmC differs substantially between the three tissue types (*Figure 6—figure supplement 2*). Consequently, we expected the association between mutation frequency and 5hmC to be highest when mutation and modification data are derived from matching tissue types. To test this hypothesis, we used a GLM on genomic windows of 100 kbp to predict CpG>T mutation rate from a combination of 5hmC_{rel} maps of all three tissues. Strikingly, for each cancer type, the best predictor came from the same tissue type (*Figure 6A*), suggesting that tissue differences in 5hmC are reflected in the CpG>T mutation landscape. The same results were obtained in all available replicates of the 5hmC_{rel} maps (*Figure 6—figure supplement 1B*). Finally, we added a 5hmC_{rel} map derived from embryonic stem cells (ESC) as an additional predictor, to compare our findings to previously reported results (*Supek et al., 2014*). As we anticipated, the ESC-derived 5hmC levels have substantially lower predictive power on CpG>T mutation rate than any of the tissue-derived maps.

While base-resolution maps of 5hmC for human tissue are still rare, there is a wide range of BS-Seq data sets available in public databases. Given our findings thus far, we predicted that tissues with high levels of 5hmC relative to 5mC would exhibit fewer CpG>T mutations in modified sites than tissues with low total 5hmC. To test this hypothesis, we measured total levels of 5mC and 5hmC using High Pressure Liquid Chromatography (HPLC-UV) in DNA of eight human tissue types for which BS-Seq maps are publicly available (*Figure 6—figure supplement 3*). In order to account for unrelated cancer-type specific differences in CpG>T mutability, we normalised the mutation frequency in modified sites by the mutation frequency in unmodified sites. The analysis of association between genomic 5hmC and enrichment of CpG>T mutations revealed a strong negative correlation (*Figure 6C*) in all tissue types except lung. Interestingly, this difference seems to stem from smoking-related effects. Lung cancer mutation data from heavy smokers revealed a markedly lower frequency of CpG>T mutations in modified sites, relative to other mutations. It has been reported that the typical C>A mutational signature associated with smoking was found significantly enriched in CpGs outside CpG islands, suggesting that it preferentially occurs at modified CpG sites (*Pleasance et al., 2010*). Accordingly, our data indicate that CpG>T mutations might also be differentially affected by smoking-related mutagens.

## Discussion

Here we have established a link between the landscape of DNA modifications and the mutational profile of somatic human cells. Our measurements indicate that 5hmCs carry between 28 and 53% fewer mutations than methylated cytosines in brain. This results in a mutational load at 5hmC sites that is comparable to that of unmodified cytosines in CpG dinucleotides. This effect is not only observable in brain, but also in kidney cancers and myeloid leukaemias. The relationship between 5hmC and CpG>T mutation rate can be detected at the scale of the exome as well as genome-wide and is independent of other region-specific influences on mutation frequency. We show that the

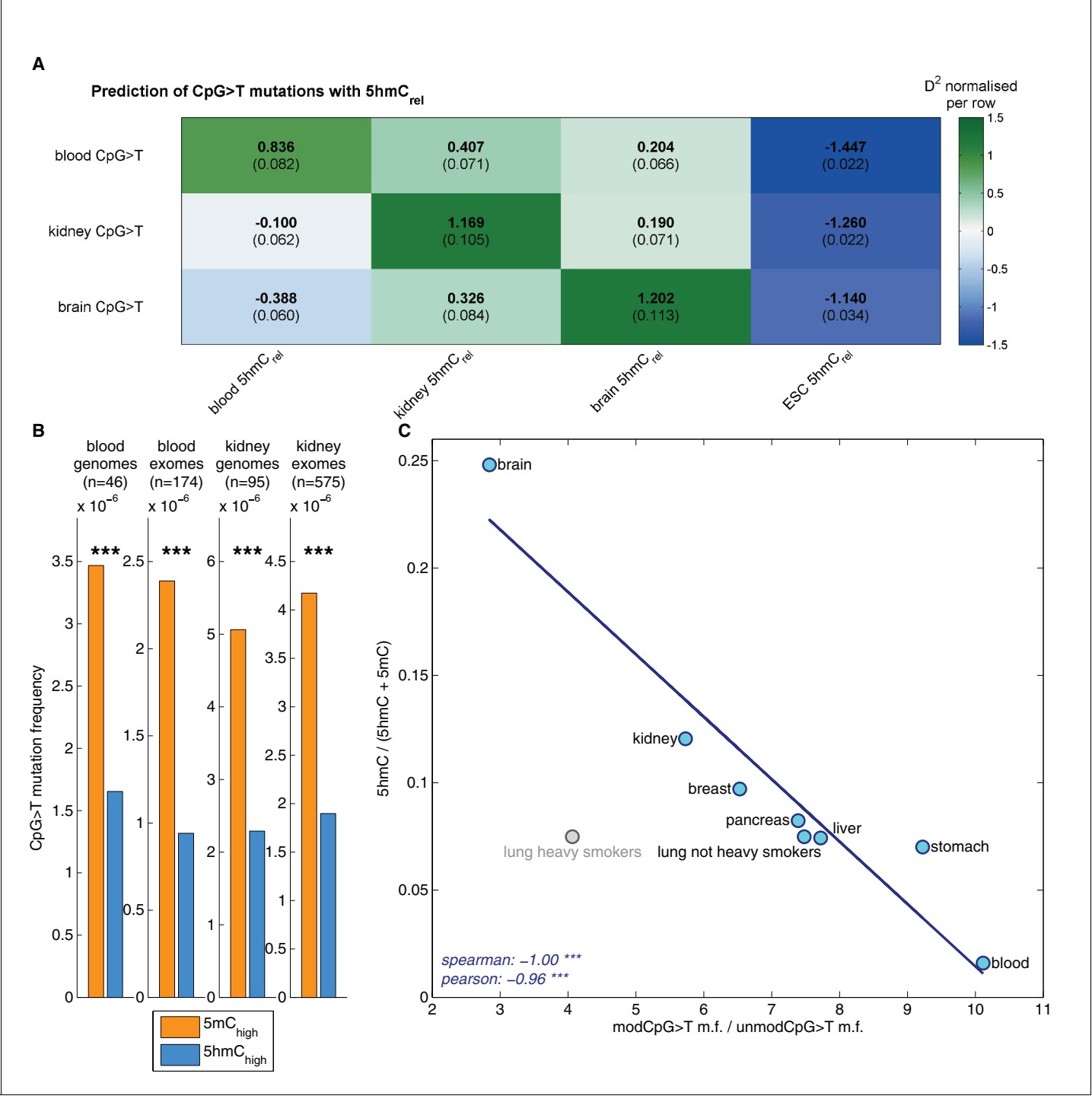

**Figure 6.** Decreased CpG>T mutation frequency in 5hmC is not limited to brain tissue. (**A**) Predictions of CpG>T mutation frequency in whole genome cancers in blood (AML), kidney and brain using 5hmC$_{rel}$ maps from blood, kidney, brain and embryonic stem cells (ESC) in 100 kbp genomic windows. The values are z-score normalised per rows in order to normalise for different number of patients and mutations in each cancer type (the original $D^2$ values are in parentheses); the higher values of $D^2$ (green), the better predictions. (**B**) CpG>T mutation frequency in 5mC vs. 5hmC in kidney and blood. (**C**) Correlation of total 5hmC$_{rel}$ levels (measured with HPLC) with frequency of CpG>T mutations in modified cytosines normalised by the frequency in unmodified cytosines in different tissues (see Materials and methods).

The following figure supplements are available for figure 6:

*Figure 6 continued on next page*

*Figure 6 continued*

**Figure supplement 1.** Decreased CpG>T mutation frequency in 5hmC is present in three tissues consistently for different replicates of modification maps.

**Figure supplement 2.** Comparison of 5hmC in 10 kbp windows in blood, kidney (2 replicates), and brain.

**Figure supplement 3.** HPLC measurements of total 5hmC and 5mC in eight tissues.

relative impact of hydroxymethylation on mutagenesis decreases proportionally to the level of 5hmC in the tissue, suggesting that it represents a general property of this DNA modification.

Two possible scenarios could explain the striking difference in mutability between 5mC and 5hmC. Firstly, spontaneous and enzymatic deamination reactions of 5hmC could be less favourable than 5mC. As a consequence, fewer new mutation events would be expected at 5hmC sites. Indeed, cytosine deaminases (namely, AID and APOBEC1-3) have 4.4–38x lower activity on sites with 5hmC compared to 5mC, supporting this possibility (*Nabel et al., 2012*; *Rangam et al., 2012*). Secondly, deamination of 5mC produces thymine whereas 5hmC deaminates to 5-hydroxymethyluracil (5hmU). This atypical base in DNA could be more efficiently recognised and replaced by the DNA glycosylases initiating base-excision repair (BER). Determining the relative contribution of DNA glycosylases to the lower mutation rate would be challenging, since some of these enzymes recognise several types of mismatches. TDG and MBD4 excise both T and 5hmU when mis-paired with G (*Hardeland et al., 2003*; *Cortellino et al., 2011*; *Guo et al., 2011*; *Hashimoto et al., 2012*; *Moréra et al., 2012*), whereas SMUG1 does not repair T:G but has a robust activity for 5hmU:G (*Nilsen et al., 2001*; *Kemmerich et al., 2012*). Therefore, there might be more efficient repair of 5hmU in the genome. Further genome sequencing efforts might identify patients with rare inactivating mutations in BER and/or mismatch-repair pathways that could be valuable for future investigations of the relationship between DNA repair and cytosine mutability.

It has previously been suggested that 5hmC levels increase the frequency of C>G mutations (*Supek et al., 2014*). As part of this analysis, only a very small (albeit statistically significant) decrease of C>T mutations in 5hmC sites in both SNPs and cancer SNVs was observed. There are two factors that could explain why we observe very different effects sizes for C>T and C>G mutations in 5hmC sites. Firstly, Supek et al. consider all sites with as little as one 5hmC read to be hydroxymethylated, whereas we require the level of 5hmC to exceed 5mC. In fact, when examining the effect of variation in these thresholds (*Figure 2—figure supplement 1A–F*), we noticed that the results for C>G fluctuate substantially across the range of tested cut-off values (see also *Figure 3—figure supplement 1E–F*). Secondly, we present evidence that tissue-specific changes in 5hmC patterns have great influence on the extent of correlation between 5hmC and mutability (*Figure 6B*). Specifically, 5hmC genomic localisation in embryonic stem cells was a poor predictor of CpG>T mutations in brain, kidney and blood, compared to the respective tissue-specific 5hmC patterns.

The best predictor of CpG>T mutations in any of the three tested tissues was the 5hmC$_{rel}$ map from the corresponding anatomical site. This provides evidence that the slow accumulation of CpG>T mutations in the pre-cancerous tissue was strongly influenced by the DNA modification landscape. However, any bulk tissue sample encompasses a mixture of different cell types. Mounting evidence suggests that solid tumours originate from a defined subset of cells within any one tissue type. For example, glioblastomas were proposed to originate from stem or progenitor cell types enriched in the subventricular zone, while medulloblastomas have mixed cells of origin (*Visvader, 2011*). Those cell types are of low abundance in normal tissue biopsies. The fact that we observe a clear inverse relationship between CpG>T mutations and the location of 5hmC in multiple tissue types suggests that the DNA modification landscape in cancer-progenitor cells is sufficiently similar to the tissue average to be informative about the mutation frequencies in cancer.

Under this assumption we predict that the impact of DNA modifications on the frequency of CpG>T mutations is likely to be bigger than measured here, since the terminally differentiated cells that make up the bulk of the tissue may have diverged further from cancer-progenitor cells. Advancements in the identification of cancer origins and isolation of single cells, combined with

single-cell bisulfite sequencing, will enable an improved assessment of the impact of DNA modifications on mutability.

The strong correlation between relative 5hmC levels in a tissue and the mutability of modified cytosine also points towards a shared underlying mutagenic process. The notable deviation of smoking-induced lung-cancers supports this hypothesis. We speculate that a yet undefined smoking-induced mutagenic mechanism preferentially affects unmethylated CpG sites. More experimental work will be needed to elucidate the biochemical causes for this phenomenon. In the future, the linear relationship between 5hmC levels and CpG>T mutation rate could thus be used to identify other environmental mutagens with a differential effect on modified cytosines.

# Materials and methods

## Code
Most of the analyses were performed using Matlab. Code and other required files are available on Figshare under doi 10.6084/m9.figshare.c.3249394 (*Tomkova et al., 2016*).

## Mutation data
Cancer somatic mutations (see *Supplementary file 1b*) were obtained from a dataset compiled by *Alexandrov et al. (2013)*, complemented with whole genome samples from ICGC, (*Wang et al., 2014*), and TCGA. Briefly, aligned reads for 49 AML tumour and normal samples were downloaded from the UCSC CGHub website under TCGA access request #10140. Somatic variants were called using Strelka (*Saunders et al., 2012*) with default parameters. All variants were classified by the pyrimidine of the mutated Watson-Crick base pair (C or T) and the immediate 5' and 3' sequence context into 96 possible mutation types as described by *Alexandrov et al. (2013)*.

## Modification data
DNA modification information (see *Supplementary file 1a*) for brain was extracted from supplementary information provided by *Wen et al. (2014)*. Only sites with more than 5 TAB-Seq reads were taken into account. $5hmC_{high}$ and $5mC_{high}$ sites were defined based on values of *mod level* (unconverted/total BS-Seq reads) and *5hmC level* (unconverted/total TAB-Seq reads) per site:

- $5mC_{high}$: *mod level* > 10% and *5hmC level / mod level* $\leq$ $threshold_{5mC}$
- $5hmC_{high}$: *mod level* > 10% and *5hmC level / mod level* $\geq$ $threshold_{5hmC}$

Effects of the choice of both thresholds were explored and then the values of $threshold_{5mC}$ = 0.3 and $threshold_{5mC}$ = 0.5 were used. In blood, BS-Seq and TAB-Seq values in CpG sites were taken from supplementary files provided by *Pacis et al. (2015)*. For kidney and ESC maps, raw reads were processed with Trim galore, Bismark (*Krueger et al., 2012*) and Mark duplicates from Picard tools. Multiple replicates were processed both independently and together (adding the reads from the replicates together). Only sites with at least 10% *mod level* were taken into account to compute $5hmC_{rel}$.

To compute the number of modified sites inside the exome, the reference Illumina Truseq definition of exon loci was downloaded from the Illumina website. Overlapping exons were merged using bedtools so that each genomic site is covered by at most one exon. Two-sided paired Wilcoxon signed-rank test was used for testing significance between mutation frequency of $5mC_{high}$ and $5hmC_{high}$ sites. The same test was used for all the following statistics, if not stated otherwise.

## Gene expression data
Gene expression (in FPKM) from RNAseq experiments on 630 brain tissue samples were downloaded from the GTEx human tissue expression project (http://www.gtexportal.org/home/).

## Visualisation on genome
The following genomic features were computed in 100 kbp windows: average 5hmC, 5mC, $5hmC_{rel}$ (all from the supplementary information provided by *Wen et al. (2014)*, i.e. mod $\geq$ 10%), average *log(1 + gene expression)*, gene density, exon density, CpG density, modCpG density, CpG island (CGI) density, and average modification level (from raw BS-Seq reads). These features and CpG>T

mutation frequency (from MDB and PA whole-genome sequencing datasets) were z-score normalised and plotted per chromosome after Gaussian smoothing with parameters n = 50, sigma = 2.5.

## Mutation frequency in highly and lowly expressed genes

Genes were sorted according to their median expression values. The upper 50-percentile (9701 genes) were classified as highly expressed, the rest as lowly expressed. Introns were included only for whole genome samples.

## Pairing of 5mC and 5hmC sites

For each $5hmC_{high}$ site in random order, the nearest not previously selected $5mC_{high}$ site was selected such that the 5mC-5hmC pair fulfilled the following conditions: both $5hmC_{high}$ and $5mC_{high}$ sites are inside an exon or both are outside exons, and both share the same context (CG, CHG, and CHH, where H is T, A or C). This resulted in 6801374 pairs with a median distance of 1 and $25^{th}$ and $75^{th}$ quantiles of -177 and +177, respectively.

## Mutation frequency around aligned 5mC and 5hmC

Modified sites with no other modifications in a 2 kb radius were selected (374000 sites with 5mC and the same number of 5hmC sites), and the mutation frequency up to 2 kbp upstream and downstream (in bins without other modifications) was plotted.

## Gradients analysis

All modified cytosines (i.e., *mod level* > 10%) in the CpG context were divided into 9 right-open intervals according to their ratio of *5hmC level* to *mod level*. The leftmost bin contained cytosines where the major modification is 5mC, while the rightmost bin contained cytosines where the major modification is 5hmC. In each bin, the frequency of mutations was computed and plotted. A linear regression model was fitted to the data (function fitlm in Matlab) and the significance of the linear coefficient was tested using F-test for the hypothesis that the regression coefficient is zero (function coefTest in Matlab). For gradients with equal binning each interval contained approximately the same number of sites (apart from the first bin, which included all values with $5hmC_{rel}=0$).

## Prediction of mutation frequency in genomic windows

CpG>T mutation frequency (response variable) and genomic features (predictors; same as above in *Visualisation on genome*) were computed in genomic windows of sizes 3 kbp–3 Mbp. Then a generalised linear model (fitglm) assuming Poisson distribution of the response variable was fitted with a linear model specification (i.e., *intercept + linear term for each predictor*) and DispersionFlag set to true. Model fits were compared in terms of $D^2$ and p-value (model.devianceTest), as recommended, e.g., in *Guisan and Zimmermann (2000)*, *Mittlböck and Heinzl (2004)*.

## Simulation of effects of number of patients on GLM

Each chromosome was split into windows of a given window size. For each window, all CpG sites were counted. A random predictor was generated in each window with a beta distribution (*Beta (3,4)*). For each patient, a random number of mutations in each window was generated as

$$Binomial((n = windowSize(iWindow), p = \frac{predictor(iWindow)}{coefficient})$$

where:

$$coefficient = \frac{\sum_{iWindow} windowSize(iWindow) * predictor(iWindow)}{174}$$

The coefficient was set so that the expected total number of mutations per patient summed to 174, the observed average number of CpG>T mutations in brain WGS data. The response variable was set as the average CpG>T mutation frequency over all patients. A GLM was fit on the given predictor and response variable and $D^2$ was measured. The process was repeated 10 times for each combination of window size and number of patients.

## Gene-wise prediction of mutation frequency

Mutation frequency was modelled with two predictor variables: average $5hmC_{rel}$ per gene and $\log_e$-transformed gene expression. The following response variables computed in exons of each gene were compared:

- modC>T: number of C>T mutations in modified C sites / number of modified C sites
- CpG>T: number of C>T mutations in CpG sites / number of CpG sites
- C>T: number of C>T mutations / number of C sites
- C>N: number of mutations from C / number of C sites
- N>N: number of mutations / number of sites
- T>N: number of mutations from T / number of T sites

Genes with missing values in at least one of the predictors and genes classified as outliers in at least one response variable were excluded from the analysis. Outliers were classified in the following way: $y \geq quantile(y, 0.999) + 2.5 * (quantile(y, 0.999) - quantile(y, 0.001))$. Out of 17,605 genes, 10 were classified and removed as outliers: ASPN, BBOX1, CCL4, ESPN, FOLH1, HLA-DPB1, IDH1, NLRP6, S100P, and TP53. The same GLM model as above was used. To calculate the relative contribution of one predictor variable over the other, two models were fitted with either one or both predictor variables and the difference in $D^2$ was used.

## HPLC measurements of total 5hmC and 5mC in eight tissues

10µg of genomic DNA (amsbio; D1234003, D1234004, D1234035, D1234086, D1234090, D1234122, D1234142, D1234148, D1234149, D1234152, D1234171, D1234188, D1234200, D1234206, D1234226, D1234227, D1234246, D1234248, D1234260, D1234274, HG-101) was treated with 1U RNase A (Thermo Scientific), purified by phenol chloroform ethanol precipitation and incubated overnight in hydrolysis solution (45 mM NaCl, 9 mM MgCl2, 9 mM Tris pH 7.9, $\geq$250 U/ml Benzonase (sigma), 50 mU/ml Phosphodiesterase I, $\geq$20 U/ml Alkaline phosphatase, 46.8 ng/ml EHNA hydrochloride, 8.64 µM deferoxamine). Protein components were removed by centrifugation through Amicon centrifugal filter unit (3 kDa cut-off, Millipore) before samples were lyophilised and resuspended in buffer A. Nucleosides were resolved with an Agilent UHPLC 1290 instrument fitted with Eclipse Plus C18 RRHD 1.8 µm (2.1 $\times$ 150 mm column) and detected and quantified with Agilent 1290 DAD fitted with a Max-Light 60 mm cell. Buffer A was 100 mM ammonium acetate, pH 6.5; buffer B was 40% acetonitrile, and the flow rate 0.4 ml min−1. The gradient was between 1.8–100% of 40% acetonitrile with the following steps: 1–2 min, 100% A; 2–16 min 98.2% A, 1.8% B; 16–18 min 70% A, 30% B; 18–20 min 50% A, 50% B; 20–21.5 min 25% A, 75% B; 21.5–22.5 min 100% B; 22.5–24.5 min 100% A. Relative abundance of 5mC and 5hmC were established by detection of adenosine at 280nm allowing determination of total cytosine by extinction coefficient calculation using standards.

## Acknowledgements

We would like to thank Jakub Tomek, Pijus Brazauskas and David Severson for helpful discussions. SK and BS-B are funded by Ludwig Cancer Research. SK received funding from BBSRC grant BB/M001873/1. MT is funded by EPSRC (EP/F500394/1) and Bakala Foundation. This study makes use of data generated by the Blueprint Consortium. A full list of the investigators who contributed to the generation of the data is available from www.blueprint-epigenome.eu. Funding for the project was provided by the European Union's Seventh Framework Programme (FP7/2007–2013) under grant agreement no 282510 – BLUEPRINT.

## Additional information

### Funding

| Funder | Grant reference number | Author |
|---|---|---|
| Virginia and D.K. Ludwig Fund for Cancer Research | | Marketa Tomkova<br>Michael McClellan<br>Skirmantas Kriaucionis<br>Benjamin Schuster-Boeckler |
| Engineering and Physical Sciences Research Council | EP/F500394/1 | Marketa Tomkova |
| Biotechnology and Biological Sciences Research Council | BB/M001873/1 | Michael McClellan<br>Skirmantas Kriaucionis |
| Bakala Foundation | | Marketa Tomkova |

The funders had no role in study design, data collection and interpretation, or the decision to submit the work for publication.

### Author contributions

MT, Analysis and interpretation of data, Drafting or revising the article; MM, Acquisition, analysis and interpretation of HPLC measurements; SK, BS-B, Conception and design, Drafting or revising the article

### Author ORCIDs

Benjamin Schuster-Boeckler, http://orcid.org/0000-0002-8892-5133

## Additional files

### Supplementary files

• Supplementary file 1. (a) Overview of BS-Seq and TAB-Seq data used to generate modification maps. (b) Overview of whole genome and exome sequencing data used for mutation information.

### Major datasets

The following previously published datasets were used:

| Author(s) | Year | Dataset title | Dataset URL | Database, license, and accessibility information |
|---|---|---|---|---|
| Alexandrov LB, Nik-Zainal S, Wedge DC, Aparicio SA, Behjati S, Biankin AV, Bignell GR, Bolli N, Borg A, Børresen-Dale AL, Boyault S, Burkhardt B, Butler AP, Caldas C, Davies HR, Desmedt C, Eils R, Eyfjörd JE, Foekens JA, Greaves M, Hosoda F, Hutter B, Ilicic T, Imbeaud S, Imielinski M, Imielinsk M, Jäger N, Jones DT, Jones D, Knappskog S, Kool M, Lakhani SR, López-Otín C, Martin S, Munshi NC, Nakamura H, Northcott PA, Pajic M, Papaemmanuil E, Paradiso A, Pearson JV, Puente XS, Raine K, Ramakrishna M, Richardson AL, Richter J, Rosenstiel P, Schlesner M, Schumacher TN, Span PN, Teague JW, Totoki Y, Tutt AN, Valdés-Mas R, van Buuren MM, van 't Veer L, Vincent-Salomon A, Waddell N, Yates LR, Zucman-Rossi J, Futreal PA, McDermott U, Lichter P, Meyerson M, Grimmond SM, Siebert R, Campo E, Shibata T, Pfister SM, Campbell PJ, Stratton MR, Stratton M | 2013 | Signatures of mutational processes in human cancer | ftp://ftp.sanger.ac.uk/pub/cancer/AlexandrovEtAl | Publicly available from the authors' website |
| Wen L, Li X, Yan L, Tan Y, Li R, Zhao Y, Wang Y, Xie J, Zhang Y, Song C, Yu M, Liu X, Zhu P, Hou Y, Guo H, Wu X, He C, Tang F, Qiao J | 2014 | Whole-genome analysis of 5-hydroxymethylcytosines and 5-methylcytosines at base resolution in human brain | http://www.ncbi.nlm.nih.gov/geo/query/acc.cgi?acc=GSE46710 | Publicly available at the NCBI Gene Expression Omnibus (accession no: GSE46710). |
| The GTEx Consortium | 2013 | GTEx Analysis V4 | http://gtexportal.org/static/datasets/gtex_analysis_v4/rna_seq_data/GTEx_Analysis_V4_RNA-seq_RNA-SeQCv1.1.8_gene_rpkm.gct.gz | Publicly available at GTEx Portal (http://gtexportal.org) |

| | | | | |
|---|---|---|---|---|
| Blueprint Epigenome Project Consortium | | Blueprint Epigenome | http://dcc.blueprint-epigenome.eu/#/experiments/ERX715127 | Data available via application to the BLUEPRINT Data Access Committee (http://www.blueprint-epigenome.eu/index.cfm?p=B5E93EE0-09E2-5736-A708817C27EF2DB7) |
| Internation Cancer Genome Consortium (ICGC) | | RENAL CELL CANCER - EU/FR | https://dcc.icgc.org/projects/RECA-EU | Publicly available at the ICGC Data Portal (project: RECA-EU) |
| | | The Cancer Genome Atlas (TCGA) | https://tcga-data.nci.nih.gov/tcga/ | LAML WGS data available via CGHub |
| Wang K, Yuen ST, Xu J, Lee SP, Yan HHN, Shi ST, Siu HC, Deng S, Chu KM, Law S, Chan KH, Chan ASY, Tsui WY, Ho SL, Chan AKW, Man JLK, Foglizzo V, Ng MK, Chan AS, Ching YP, Cheng GHW, Xie T, Fernandez J, Li VSW, Clevers H, Rejto PA, Mao M, Leung SY | 2014 | Whole-genome sequencing and comprehensive molecular profiling identify new driver mutations in gastric cancer | http://www.nature.com/ng/journal/v46/n6/full/ng.2983.html#supplementary-information | Publicly available at SY Leung's Laboratory website (http://web.hku.hk/~suetyi/) |
| Chen K, Zhang J, Guo Z, Ma Q, Xu Z, Zhou Y, Li Z, Liu Y, Ye X, Li X, Yuan B, Ke Y, He C, Zhou L, Liu J, Ci W. | 2015 | Loss of 5-hydroxymethylcytosine is linked to gene body hypermethylation in kidney cancer | http://www.ncbi.nlm.nih.gov/geo/query/acc.cgi?acc=GSE63183 | Publicly available at the NCBI Gene Expression Omnibus (accession no: GSE63183) |
| Roadmap Epigenomics Consortium | 2013 | Roadmap Epigenome (Breast) | http://www.ncbi.nlm.nih.gov/geo/query/acc.cgi?acc=GSM1127125 | Publicly available at the NCBI Gene Expression Omnibus (accession no: GSM1127125) |
| Roadmap Epigenomics Consortium | 2013 | Roadmap Epigenome (Pancreas) | http://www.ncbi.nlm.nih.gov/geo/query/acc.cgi?acc=GSM983651 | Publicly available at the NCBI Gene Expression Omnibus (accession no: GSM983651) |
| Roadmap Epigenomics Consortium | 2013 | Roadmap Epigenome (Lung) | http://www.ncbi.nlm.nih.gov/geo/query/acc.cgi?acc=GSM983647 | Publicly available at the NCBI Gene Expression Omnibus (accession no: GSM983647) |
| Roadmap Epigenomics Consortium | 2012 | Roadmap Epigenome (Liver) | http://www.ncbi.nlm.nih.gov/geo/query/acc.cgi?acc=GSM916049 | Publicly available at NCBI Gene Expression Omnibus (accession no: GSM916049) |
| Roadmap Epigenomics Consortium | 2013 | Roadmap Epigenome (Stomach) | http://www.ncbi.nlm.nih.gov/geo/query/acc.cgi?acc=GSM1010984 | Publicly available at NCBI Gene Expression Omnibus (accession no: GSM1010984) |

| | | | | |
|---|---|---|---|---|
| Pacis A, Tailleux L, Morin AM, Lambourne J, Maclsaac JL, Yotova V, Dumaine A, Danckaert A, Luca F, Grenier JC, Hansen KD, Gicquel B, Yu M, Pai A, He C, Tung J, Pastinen T, Kobor MS, Pique-Regi R, Gilad Y, Barreiro LB | 2015 | Bacterial infection remodels the DNA methylation landscape of human dendritic cells (TAB-seq) | http://www.ncbi.nlm.nih.gov/geo/query/acc.cgi?acc=GSE64181 | Publicly available at NCBI Gene Expression Omnibus (accession no: GSE64181) |
| Pacis A, Tailleux L, Morin AM, Lambourne J, Maclsaac JL, Yotova V, Dumaine A, Danckaert A, Luca F, Grenier JC, Hansen KD, Gicquel B, Yu M, Pai A, He C, Tung J, Pastinen T, Kobor MS, Pique-Regi R, Gilad Y, Barreiro LB | 2015 | Bacterial infection remodels the DNA methylation landscape of human dendritic cells (BS-seq) | http://www.ncbi.nlm.nih.gov/geo/query/acc.cgi?acc=GSE64177 | Publicly available at NCBI Gene Expression Omnibus under (accession no: GSE64177) |

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
