## [Decision Letter]

[Editors’ note: this article was originally rejected after discussions between the reviewers, but the paper was accepted for publication after the authors resubmitted for further consideration.]

Thank you for choosing to send your work entitled "5-hydroxymethylcytosine marks regions with reduced mutation frequency" for consideration at *eLife*. Your full submission has been evaluated by Diethard Tautz (Senior editor) and three peer reviewers, one of whom is a member of our Board of Reviewing Editors, and the decision was reached after discussions between the reviewers.

Based on our discussions and the individual reviews below, we felt that your manuscript requires major alterations that are unlikely to be accomplished within the time frame typically allotted for revised *eLife* manuscripts. We therefore regret to inform you that your work, in its present form, will not be considered further for publication in *eLife*. However, we would be happy to consider a new submission if you can demonstrate that 5hmC causes a strong and general reduction of C>T mutations associated with 5mC.

*Reviewer #1:*

The main claim of the manuscript is that hydroxylation of methylcytosine (hmC) lowers the C>T transition rate of methylated cytosine (mC) in brain (cancer) cells approximately 2-fold. The authors support this claim by re-analyzing published data for the localization of DNA hydroxymethylation (BS/TAB-Seq in normal brain tissue, Wen et al., Genome Biol 2014) and for substitution rates (inferred from brain cancers, Alexandrov et al., Nature 2013). The phenomenon of less elevated transition rates in lineages leading to cancer at hydroxymethylated bases in normal brain cells is somewhat supported, but support would be bolstered by additional analyses.

Overall, the paper appears methodologically sound, but I am concerned by discrepancies with published data, and the biology doesn't quite add up. A paper published last year and cited here (Supek at al., PLoS Genetics 2014) reported elevated levels of C>G transversions associated with hmC in human and mouse. The same paper also found modest but significant reduction of C>T transitions in both species, interpreted as an expected outcome of the chemical differences between hmC and mC. This important result isn't mentioned by the authors, who report a much greater reduction – the main novel finding of this paper. Unlike the published result, the authors' analysis relies on a single human brain hmC dataset, and the substitution rates in cancer lineages are not obviously matched to bulk modification levels of an individual brain. Although the authors claim the matching of datasets is a strength of their analysis, it is actually somewhat of a weakness because the samples are not directly comparable.

It is noted that "all brain cancer types individually displayed a significant (28-53%) reduction of C>T mutations in 5hmChigh sites (Figure 1), making it highly improbable that the observation is an artefact of tissue type or sequencing method", however a similar result across the board makes it suspect of a systematic artifact, potentially caused by reliance on a single hmC dataset. I think it is very important for the authors to perform their analyses with additional hmC datasets.

I find the author's functional claims problematic. hmC modification decreases in most cancer lineages measured (Ficz and Gribben, Genomics 2014), so how can hmC continue to lower mutation rates if it is increasingly lost in the proliferating cancer cells? Even if it were only acting early in cancer development when hmC would presumably still be high, wouldn't its mutational signature be quickly overwhelmed by subsequent mutations occurring on non-hydroxymethylated cytosines? Furthermore, the most mitotically active cells, neuronal progenitors and neural stem cells, have the least hmC (Wen and Tang, Genomics 2014) – yet, aren't these thought to be more likely to give rise to the cancer lineages?

More generally, the argument that elevated hmC protects long-lived cells like neurons is odd, because these cells are not in obvious need of special protection from mutations. Mutations have the greatest potential for harm when arising in cells that give rise to many other cells. Primary adult brain tumors overwhelmingly arise from glial cells. Elevated levels of hmC in long-lived cells would seem to argue against a function in reducing mutations, or at least not for such a function.

*Reviewer #3:*

General Assessment

The authors analyze the relationship between the distribution of hydroxymethylation and mutational patterns in brain cancer exomes and whole genomes. Authors make a nice effort to collect a large sample size of brain cancer samples. The paper is well written and I do not see any major methodological pitfall in most of their analyses. However, I am skeptical about the generality about their findings, and also feel that some of their claims should be toned down or strengthened by additional analyses.

1) The title and some of the Discussion indicates that the findings in this paper may represent a universal mutational trend. However, their findings clearly cannot be generalized to all somatic mutations. In this respect the title should be changed to specify that the patterns were observed in brain cancers.

Specifically, if 5hmC is intrinsically linked to lower C>T mutability, this pattern should be also observed in other cancer types as well as in non-cancer context. As cited in this paper, reduced C>T pattern was not consistently observed in a previous study (Supek et al.) using population variation and SNPs from several cancer types. Authors mention that this discrepancy is due to specific 5hmC patterns of each tissue. However, to claim that C>T mutation is universal authors need to discard specific mutational biases in brain cancer. For example, one might come up with an alternative explanation that the repair machineries for 5mC mismatch is specifically impaired in brain cancers, and consequently they are highly mutable in these specific cell types. A scenario such as this can be easily tested by comparing expression patterns of genes involved in 5mC and 5hmC repair machineries for different cell types. Second, it might be necessary to confirm that the discrepancies are not because of different methodological approaches. Authors could follow 5hmC – 5mC context matching procedure from the previous study and/or analyze population-based SNP data themselves. Finally, even though germline 5hmC maps are not yet available, one can infer those patterns using the existing maps and using the principles of parsimony. For example, the cytosines that are hydroxymethylated in both ESC and a differentiated tissue such as cortex may be inferred to be hydroxymethylated in germline and should associate with less C>T SNP at the population level (having a larger number of 5hmC maps would increase the confidence of the inference). In contrast, brain-specific 5hmC (5mC in ESC) are expected to show less C>T mutations in brain cancers, but more C>T at population-level (since these positions are expected to be 5mC in germlines).

2) In addition, it is stated in the Abstract that the levels of 5hmC have 'predictive' power for mutation frequencies, in particular non-synonymous mutation frequencies. However, without the total R2 of the model it is impossible to judge the predictive power of their models. Moreover, the differences in R2 of models including or excluding 5hmC levels are so small (in the other of 1/1000th, per Figure 4), it is a stretch to state that the '5hmC levels are predictive of lower non-synonymous mutation frequency' (as in the Abstract).

3) The authors conjecture that a potential (additional) role of hydroxymethylation may be to 'protect' some cell types from mutations. The authors relate to the abundance of hydroxymethylation in long-lived neurons and such a potential role. However, as authors acknowledge in the Discussion, this must be taken with caution since the biological significance as well as tissue-wide distribution of hydroxymethylation is still poorly understood. Authors might also acknowledge that an additional caveat of this work is that the sample studied is heterogeneous. The frontal cortex is composed of not only long-lived neurons but also shorter-lived glial cells. To directly test their hypothesis, it might be necessary to compare sorted cell types with close developmental origin and biological function but with different division rates.

4) Regarding the analysis on driver genes, authors show that cancer driver genes show more 5hmC/total methylated than non-drivers genes. I feel this analysis alone is not sufficient to delineate the relationship between 5hmC and occurrence of cancer driver mutations.

The comparison between drivers and non-drivers might not be fair, since they might show different mutation rates in cancer. To test the role of 5hmC in genome stability and progression in cancer, would not it be more relevant to test if drivers are enriched for 5hmC compared to passenger genes (i.e., genes that also accumulate mutations in cancer, while not as drivers)?

If the presence of 5hmC is linked to low mutability at key genes, a testable prediction using ESC 5hmC map would be to show that the genes that are active in development show enrichment for 5hmC.

---

## [Author Response]

[Editors’ note: the author responses to the first round of peer review follow.]

Reviewer #1:

The main claim of the manuscript is that hydroxylation of methylcytosine (hmC) lowers the C>T transition rate of methylated cytosine (mC) in brain (cancer) cells approximately 2-fold. The authors support this claim by re-analyzing published data for the localization of DNA hydroxymethylation (BS/TAB-Seq in normal brain tissue, Wen et al., Genome Biol 2014) and for substitution rates (inferred from brain cancers, Alexandrov et al., Nature 2013). The phenomenon of less elevated transition rates in lineages leading to cancer at hydroxymethylated bases in normal brain cells is somewhat supported, but support would be bolstered by additional analyses.

*Overall, the paper appears methodologically sound, but I am concerned by discrepancies with published data, and the biology doesn't quite add up. A paper published last year and cited here (Supek at al., PLoS Genetics 2014) reported elevated levels of C>G transversions associated with hmC in human and mouse. The same paper also found modest but significant reduction of C>T transitions in both species, interpreted as an expected outcome of the chemical differences between hmC and mC. This important result isn't mentioned by the authors, who report a much greater reduction – the main novel finding of this paper.*

We have modified the manuscript to make the comparison between our data and Supek et al. more explicit (see Discussion, third paragraph). We now identify and discuss the likely reason for the differences between our observations and Supek et al. In particular, we show that differences in thresholds defining 5hmC sites have a major influence on the correlation with C>G mutations (Figure 2—figure supplement 1, Figure 3—figure supplement 1), and that embryonic stem cell 5hmC maps (which were used by Supek et al.) are significantly less correlated with somatic mutations than tissue-matched maps (Figure 6).

Unlike the published result, the authors' analysis relies on a single human brain hmC dataset, and the substitution rates in cancer lineages are not obviously matched to bulk modification levels of an individual brain. Although the authors claim the matching of datasets is a strength of their analysis, it is actually somewhat of a weakness because the samples are not directly comparable.

We expanded our analysis to incorporate two additional recently published 5hmC maps from kidney (2 samples) and myeloid cells (1 sample). These additional data sets also show a clear tissue-specific correlation with CpG>T mutations in corresponding cancer types (Figure 6). The best explanation for this observation is that the majority of these mutations accumulated in “normal” tissue before the onset of carcinogenesis. A significant correlation of CpG>T mutations with age supports this interpretation (Figure 2).

It is noted that "all brain cancer types individually displayed a significant (28-53%) reduction of C>T mutations in 5hmChigh sites (Figure 1), making it highly improbable that the observation is an artefact of tissue type or sequencing method", however a similar result across the board makes it suspect of a systematic artifact, potentially caused by reliance on a single hmC dataset. I think it is very important for the authors to perform their analyses with additional hmC datasets.

As mentioned above, we have included further base-resolution maps of 5hmC occupancy from two additional tissues (with several replicates). We have also performed High Pressure Liquid Chromatography (HPLC-UV) in DNA of eight human tissue types to accurately measure total 5mC and 5hmC levels. All data independently validate our hypothesis that higher 5hmC frequency reduces the likelihood of CpG>T mutations (Figure 6).

I find the author's functional claims problematic. hmC modification decreases in most cancer lineages measured (Ficz and Gribben, Genomics 2014), so how can hmC continue to lower mutation rates if it is increasingly lost in the proliferating cancer cells? Even if it were only acting early in cancer development when hmC would presumably still be high, wouldn't its mutational signature be quickly overwhelmed by subsequent mutations occurring on non-hydroxymethylated cytosines?

In order to address this criticism, we measured the relationship between age of diagnosis in patients and the number of CpG>T mutations. If the mutational signature was dominated by CpG>T mutations that occurred *during* cancer growth, we would expect little correlation with patient age. However, we found good correlation between age and CpG>T mutations in modified sites (Figure 2). Moreover, mutations at 5hmC sites accumulated slower than at 5mC sites. This supports the argument that the majority of CpG>T mutations occurred before the onset of tumorigenesis (i.e. before any cancer-associated decrease in 5hmC). Clock-like kinetics of CpG>T mutations was observed by others, suggesting that this is a universal mutational process occurring in normal somatic cells (Alexandrov et al. 2015, Nat Genet.). We have amended the manuscript with further references on this question (subsection “5hmC sites in brain exhibit lower frequency of CpG>T mutations than 5mC sites”).

Furthermore, the most mitotically active cells, neuronal progenitors and neural stem cells, have the least hmC (Wen and Tang, Genomics 2014) – yet, aren't these thought to be more likely to give rise to the cancer lineages?

More generally, the argument that elevated hmC protects long-lived cells like neurons is odd, because these cells are not in obvious need of special protection from mutations. Mutations have the greatest potential for harm when arising in cells that give rise to many other cells. Primary adult brain tumors overwhelmingly arise from glial cells. Elevated levels of hmC in long-lived cells would seem to argue against a function in reducing mutations, or at least not for such a function.

We carefully considered the reviewers comments and decided that more data will be needed to address the question whether 5hmC levels are themselves under selection. We removed this part of the analysis from the manuscript, and instead focused more on the core finding of our paper, namely that 5hmC has a substantial effect on CpG mutagenesis across tissue types.

Reviewer #3:

General Assessment

The authors analyze the relationship between the distribution of hydroxymethylation and mutational patterns in brain cancer exomes and whole genomes. Authors make a nice effort to collect a large sample size of brain cancer samples. The paper is well written and I do not see any major methodological pitfall in most of their analyses. However, I am skeptical about the generality about their findings, and also feel that some of their claims should be toned down or strengthened by additional analyses.

1) The title and some of the Discussion indicates that the findings in this paper may represent a universal mutational trend. However, their findings clearly cannot be generalized to all somatic mutations. In this respect the title should be changed to specify that the patterns were observed in brain cancers.

As mentioned in reply to reviewer 1, we expanded our analysis to incorporate two additional recently published 5hmC maps from kidney and myeloid cells. These additional data sets show a clear tissue-specific correlation with CpG>T mutations in corresponding cancer types (Figure 6—figure supplement 9). Furthermore, we performed High Pressure Liquid Chromatography (HPLC-UV) in DNA of eight human tissue types to accurately measure total 5mC and 5hmC levels (Figure 6—figure supplement 11). This enabled us to chart the relationship between genome-wide mutation rate and relative levels of cytosine modifications in a wide range of tissues. All data independently validate our hypothesis that higher 5hmC occupancy reduces the likelihood of CpG>T mutations. We hope that this evidence will confirm that our findings are indeed generalisable.

Specifically, if 5hmC is intrinsically linked to lower C>T mutability, this pattern should be also observed in other cancer types as well as in non-cancer context. As cited in this paper, reduced C>T pattern was not consistently observed in a previous study (Supek et al.) using population variation and SNPs from several cancer types. Authors mention that this discrepancy is due to specific 5hmC patterns of each tissue. However, to claim that C>T mutation is universal authors need to discard specific mutational biases in brain cancer. For example, one might come up with an alternative explanation that the repair machineries for 5mC mismatch is specifically impaired in brain cancers, and consequently they are highly mutable in these specific cell types. A scenario such as this can be easily tested by comparing expression patterns of genes involved in 5mC and 5hmC repair machineries for different cell types.

Mutational signatures vary greatly between different brain cancers, kidney cancer and leukaemia, yet we observe a correlation between 5hmC and CpG>T mutagenesis in our expanded analysis of additional 5hmC maps from all these sites (Figure 6). We conclude that it is highly unlikely that a tissue artefact explains the reduced mutation rate in 5hmC sites.

Second, it might be necessary to confirm that the discrepancies are not because of different methodological approaches. Authors could follow 5hmC – 5mC context matching procedure from the previous study and/or analyze population-based SNP data themselves.

We expanded the comparison of our results with those of Supek et al. In particular, we show that differences in thresholds defining which sites are considered 5mC and 5hmC, respectively, have a major influence on the correlation with C>G mutations. We found that when applying thresholds as reported by Supek et al., we can confirm an increase in C>G mutations at “5hmC” sites. However, we believe that the thresholds we applied are a more conservative choice, as we discuss in the revised manuscript.

Furthermore, we also show that 5hmC maps from embryonic stem cells used by Supek et al. are significantly less correlated with somatic mutations than tissue-matched maps. This explains the difference in magnitude of CpG>T effect reported in their paper.

Finally, even though germline 5hmC maps are not yet available, one can infer those patterns using the existing maps and using the principles of parsimony. For example, the cytosines that are hydroxymethylated in both ESC and a differentiated tissue such as cortex may be inferred to be hydroxymethylated in germline and should associate with less C>T SNP at the population level (having a larger number of 5hmC maps would increase the confidence of the inference). In contrast, brain-specific 5hmC (5mC in ESC) are expected to show less C>T mutations in brain cancers, but more C>T at population-level (since these positions are expected to be 5mC in germlines).

We explored this possibility, but found that the overlap between 5hmC maps from brain and embryonic stem cells is very small. Out of 6,501,153 loci that qualify as 5hmC_high_ in brain, only 78,789 (1.2%) overlap with ESC 5hmC_high_ loci. The overlap between all four maps was even smaller, containing only 296 5hmC_high_ loci. As a result, the diminished statistical power restricts the ability to observe any effects. However, our new analysis on maps from three tissues addresses the reviewer’s point that ‘a larger number of 5hmC maps would increase confidence’. In fact, we show a clear tissue-specific correlation with CpG>T mutations in the corresponding cancer types.

2) In addition, it is stated in the Abstract that the levels of 5hmC have 'predictive' power for mutation frequencies, in particular non-synonymous mutation frequencies. However, without the total R2 of the model it is impossible to judge the predictive power of their models. Moreover, the differences in R2 of models including or excluding 5hmC levels are so small (in the other of 1/1000th, per Figure 4), it is a stretch to state that the '5hmC levels are predictive of lower non-synonymous mutation frequency' (as in the Abstract).

The Abstract now does not refer to the predictive power of 5hmC anymore. In general, we have updated this analysis in several aspects. Firstly, we now use the more appropriate D^2^ measure (instead of R^2^), since we use *generalised* linear models (GLM). Secondly, we simulated data to test how the D^2^ of a “perfect” predictor of mutation rate would be affected by different window sizes and patient numbers. We show that for the given number of patients and the specific window size that we use in our analysis, the D^2^ is *expected* to be relatively low (Figure 5—figure supplement 1). In both our simulated data and additional GLM analyses on genomic windows, the values of D^2^ increase with larger sized windows (Figure 5, Figure 5—figure supplement 1). For instance, in 3Mbp windows in the real data, the D^2^ is as high as 0.45 for a single predictor (Figure 5).

3) The authors conjecture that a potential (additional) role of hydroxymethylation may be to 'protect' some cell types from mutations. The authors relate to the abundance of hydroxymethylation in long-lived neurons and such a potential role. However, as authors acknowledge in the Discussion, this must be taken with caution since the biological significance as well as tissue-wide distribution of hydroxymethylation is still poorly understood. Authors might also acknowledge that an additional caveat of this work is that the sample studied is heterogeneous. The frontal cortex is composed of not only long-lived neurons but also shorter-lived glial cells. To directly test their hypothesis, it might be necessary to compare sorted cell types with close developmental origin and biological function but with different division rates.

We have carefully considered this point and agree with the reviewer that more data will be needed to address this particular question adequately. Hence, we decided to remove this tangential part of the analysis and instead focus on the central point of 5hmC mutation rate across cell types. Please see also the response to Reviewer 1 regarding neuronal cells.

4) Regarding the analysis on driver genes, authors show that cancer driver genes show more 5hmC/total methylated than non-drivers genes. I feel this analysis alone is not sufficient to delineate the relationship between 5hmC and occurrence of cancer driver mutations.

The comparison between drivers and non-drivers might not be fair, since they might show different mutation rates in cancer. To test the role of 5hmC in genome stability and progression in cancer, would not it be more relevant to test if drivers are enriched for 5hmC compared to passenger genes (i.e., genes that also accumulate mutations in cancer, while not as drivers)?

If the presence of 5hmC is linked to low mutability at key genes, a testable prediction using ESC 5hmC map would be to show that the genes that are active in development show enrichment for 5hmC.

This paragraph has been removed as the manuscript now focuses on more in-depth analysis of 5hmC mutation rate across tissue types.